# The Influence of the Schoolbag on Standing Posture of First-Year Elementary School Students

**DOI:** 10.3390/ijerph16203946

**Published:** 2019-10-16

**Authors:** Ivan Jurak, Ozren Rađenović, Filip Bolčević, Andreja Bartolac, Vladimir Medved

**Affiliations:** 1Department of Physiotherapy, University of Applied Health Sciences, 10000 Zagreb, Croatia; ivan.jurak@zvu.hr; 2Department of Occupational Therapy, University of Applied Health Sciences, 10000 Zagreb, Croatia; ozren.radenovic@zvu.hr (O.R.); andreja.bartolac@zvu.hr (A.B.); 3Faculty of Kinesiology, University of Zagreb, 10000 Zagreb, Croatia; filip.bolcevic@kif.hr

**Keywords:** center of gravity shift, anterior pelvic tilt angle, craniovertebral angle, craniocervical angle, kinematics

## Abstract

The purpose of this study was to determine the influence of the mass of a schoolbag on standing posture in first-year elementary school children. First-year elementary school students (*n* = 76) participated in this study. The data was digitized and analyzed using SkillSpector and Kinovea. Results have shown a change of Center of Gravity (COG) position in all three anatomical planes (*p* < 0.01), as well as a change in two out of three measured postural angles—craniovertebral (*p* < 0.01) and craniocervical (*p* < 0.01) angle. The most important aspect of changed posture, anterior shift of COG, was measured to be 2.4 cm and was in moderate negative correlation with student body mass (−0.4, *p* < 0.01) and height (−0.4, *p* < 0.01) when students were encumbered with a schoolbag weighing 16.11% of their body mass, on the average. Also, this study confirms that when encumbered, students’ head posture shifts to a more protracted position.

## 1. Introduction

A schoolbag is a necessary item for any student. The general public opinion is that schoolbags are too heavy, but science has yet to empirically establish the hazards of carrying a schoolbag to the health of children. The “safe” upper limit of schoolbag mass needs to be determined, as does the relationship between the mass of a schoolbag and any potential musculoskeletal disorders. Studies dealing with demographic data and the ratio of child mass and schoolbag mass mostly state that the average ratio of child mass and schoolbag mass is 10% to 15% or more [1,2,3,4].

American Academy of Orthopedic Surgeons recommend that the mass of the schoolbag should not exceed 15% of the child’s mass [5], while the American Association of Occupational Therapists recommends the mass of the schoolbag not be above 10% [6]. The generally recommended ratio of schoolbag to student mass is between 10% and 15%, but appropriate postural adaptation occurs at loads as small as 3% [7]. Opinions are based on data, and while the influence of schoolbags on health has been thoroughly researched, definite conclusions are still out of reach.

Additional recommendations for further research include standardization of the methodology and improving the quality and strength of the evidence. Since there are no hard conclusions from systematic reviews, it is important to “step back” and review biomechanical studies concerning Center of Gravity (COG) shift [8]. In this paper, the focus is on COG shift as the main measure of influence of schoolbag on child’s body posture, but also on two postural angles of the head—craniovertebral and craniocervical angle—as well as on one pelvic postural angle—anterior pelvic tilt angle. These three postural angles were chosen because research has shown that children encumbered with schoolbags usually complained of neck pain, as well as lower back pain, although to a lesser degree [9,10].

A study from 2005 [11] showed (n = 40; mean age = 12.4) that the average anterior center of mass (COM) shift was 3.4 cm when encumbered with schoolbags weighing 20% of body mass. Wearing a schoolbag in standing position resulted in anterior movement of the head and shoulders, which is not surprising given the anterior COG shift.

A study by Chansirinukor et al. [12] analyzed the changes in the cranio-horizontal angle, cranio-vertebral angle, sagittal shoulder postural angle, and lateral head inclination angle in a sample of 13 subjects and found that there were significant changes in the cervical spine posture when carrying a load amounting to 15% of the body mass, i.e., that head protraction occurs (a change of craniovertebral angle by $\overset{-}{x}$ = 1.2°). Increased head protraction when carrying 15% of body mass has also been confirmed by Ramparsad et al. [13].

In a similar study by Brackley et al. [14], the authors measured the angle of body inclination, craniovertebral angle and the angle of lumbar lordosis in 15 subjects. The results indicated a significant change in the craniovertebral angle and body inclination regardless of the way the schoolbag was carried (high, middle, or low position of the bag on the back). The study also states that although the postural changes were evident, the study subjects did not notice any pain or extra strain, so the authors concluded that it was impossible to conclude with absolute certainty that the maximum recommended ratio of 15% of mass is too high. 

Even though the influence of schoolbags on student health is a debated topic, a review of the literature shows that there is no clear consensus on how mass and placement of schoolbag influence postural adaptation. Both Steele et al. [8] and Abdullah et al. [15] agree and suggest that future research requires a more rigorous approach, with valid and reliable measuring instruments as well as appropriate sample sizes. In this particular study, we aim to measure how encumbrance with school bag influences COG movement and to identify the extent of cervical and pelvic postural adaptations using widely available and relatively accessible methods, which should be very easy to replicate, therefore making findings comparable.

The overall aim of this study is to research the correlation of schoolbag mass and standing posture in first-year elementary school children.

The specific objectives are:To measure the shift of center of gravity (cm) in first-grade primary school students along the sagittal, coronal, and vertical axis while encumbered with a schoolbag.To measure the change in the anterior pelvic tilt (APT) angle, craniovertebral angle (CV) and craniocervical (CC) angle while encumbered with a schoolbag.To calculate the correlation between the shift of center of gravity and body height, mass, BMI (Body Mass Index) and the anterior pelvic tilt (APT) angle, craniovertebral angle (CV) and craniocervical angle (CC).To ascertain the difference in the shift of center of gravity (cm) along the sagittal, coronal, and vertical axis between boys and girls.

## 2. Materials and Methods 

### 2.1. Subjects

The study was conducted on a sample of 76 first-grade students (35 boys and 41 girls) in a convenience sample recruited in two primary schools in Zagreb, Croatia. First-year elementary school students in Croatia are universally seven years old so specific information about age was not collected.

### 2.2. Procedure

The study was conducted using a parent questionnaire and some anthropologic measurement methods including kinematic measurements appropriate for children. The questionnaire provided the researchers with data on the gender and schoolbag carrying habits of the children. Students’ height and mass were collected. This study was conducted respecting the International Charter for Ethical Research Involving Children and we obtained ethics approval from the University of Applied Health Sciences Ethics Committee (approval no. 602-04/19-18/671). Also, researchers were required to obtain informed consent from parents or legally appointed guardians. The children were introduced to the measurement methods to be used and they were not exposed to any health risks. Since the measurement process was conducted on school premises, permission was also obtained from both school headmistresses. 

All measurements were taken on the school premises in the school sports hall at 08:00, before the beginning of lessons. Complete anthropometric and kinematic measurements took about 120 minutes overall (about 10 minutes per child). The measurements were taken by six researchers/research assistants in three teams. The children were dressed for physical education class and unshod but wearing socks. Anthropometric measurements were taken using a Harpenden anthropometer while body mass and mass of schoolbags was measured using ground reaction measurement platform Kistler Quattro Jump, Type 9290AD (Kistler Group, Winterthur, Switzerland) of high accuracy. Video signals for kinematic measurements were captured using a Pentax K20D digital camera (Pentax, Tokyo, Japan) mounted on a tripod. Before any kinematics were recorded, the average mass of schoolbags was determined by weighing all of the participants’ schoolbags. All other measurements were taken in the same order—anthropometrics, body mass and kinematics. During video signal capture, students were instructed to stand at predetermined positions and conditions (without/with schoolbag). The same school bag with predetermined average mass was used for every subject. Straps were adjusted for each subject based on subjects’ perception of comfort. Participants were always recorded in the same order with no randomization.

### 2.3. Anthropometrics

Measurements of the students’ height and mass, and the mass of the students’ full schoolbags were taken. Measurements were also taken of the relative distance between the spina illiaca anterior superior (SIAS) and the spina illiaca posterior superior (SIPS), and the heights of the SIAS and SIPS, to ascertain the anterior pelvic tilt angle according to Sanders and Stavrakas [16]. With those measurements acquired, the sine of the APT angle was calculated using trigonometric equations. This method has been shown to have good intratester reliability coefficients for test-retest measurement of 0.88 [15]. The APT angle was measured while the child was carrying a schoolbag and without it, to observe the influence of the schoolbag mass on the position of the pelvis and the lumbar spine. 

### 2.4. Kinematics

Change of center of gravity, craniovertebral and craniocervical angles were the kinematic quantities sought. A digital camera was used, placed on a standardized tripod 12 meters away from the subject, in order to capture a full body view of the posture. Using photo-reflective material, the C7 spinous process was marked on the subject, as well as the SIAS point, bilaterally. Marking of the C7 spinous process was required in order to calculate the CV and CC angles of the neck, while the marked SIAS points enabled easier data digitalization later on during image processing. The subject then placed his or her heels at points previously marked on the floor in front of a monochromatic background, for enhanced contrast.

#### 2.4.1. Center of Gravity

In processing raw data, we used SkillSpector software, ver. 1.3.2. (Video4coach, Odense, Denmark), designed to analyze motion in 2D or 3D space. SkillSpector (SS) is one of many popular, freeware, motion capture software packages that uses photogrammetric methods to analyze movement. It is capable of calculating body segment angles, movement of center of gravity, linear and angular kinematics. It is less accurate than standard optoelectronic systems, but it is more readily available and more appropriate to use outside the laboratory. The accuracy of such software depends on many factors (camera, perspective, calibration, human factors), so recommended procedures must be followed [17]. When used appropriately, the accuracy of such systems has been shown to reach even the 0.3 mm margin [18], and the interrater reliability was moderate to high, with intraclass correlation coefficients (ICC) ranging from 0.71 to 0.99 [19]. 

The digital model used for the antero-posterior (AP) projection is called Simple Full Body and consists of 18 points, including 16 bilateral points—akropodion, sphyrion fibulare, tibiale laterale, SIAS, acromion, radiale, stylion, dactylion, and 2 points along the sagittal line—gnathion and the center of os frontale. The Full Body Right Side digital model was used for the latero-lateral (LL) projection (right profile), which consists of 10 points—akropodion, sphyrion fibulare, tibiale laterale, SIAS, akromion, radiale, stylion, dactylion, gnathion and the center of os frontale. The subjects were photographed in two projections using one camera and a 2D analysis of the AP and LL views independently of each other was conducted. For calibration two-dimensional postural calibration option was used.

#### 2.4.2. Craniovertebral Angle

The CV angle is the angle between the imaginary line connecting the ear tragus and the spinous process of C7, and the horizontal plane (Figure 1), used to obtain data on the degree of flexion of the head and neck. The key factor in the analysis of camera image to obtain the CV angle was marking the anthropometric point C7 with photo-reflective material to gain precise measurements. As the tragus anthropometric point is very visible, there was no need for marking.

#### 2.4.3. Craniocervical Angle

The CC angle is the angle between the imaginary line connecting the ear tragus and C7 vertebra and a line connecting the tragus of the ear and canthus of the eye [20] (Figure 2), and is used to gather data on the degree of head protraction. As the CC angle also uses the C7 spinous process as a reference, it was marked, while the tragus and canthus were visible enough and required no additional marking.

Both CV and CC angles were measured in the LL projection with and without the schoolbag using Kinovea software, ver. 0.8.18 (Kinovea open source project). Kinovea is popular analysis software that has excellent intrarater (ICC 0.99) and interrater (ICC 0.99) reliability [21] and has been repeatedly used as a tool for head posture angle analysis [22]. 

### 2.5. Statistical Analysis

Data were tested for normality of distribution using the Shapiro–Wilk test. Since data was found to be normally distributed, COG change and differences in CV angle, CC angle and APT angle were analyzed using paired-samples *t*–test (*p*-value was set to 0.05). Pearson’s correlations were used to test linear association between anthropometric kinematics and COG shift. For testing differences of COG shifts and CV angle, and the CC angle between boys and girls, independent samples *t*-test was used.

## 3. Results

### 3.1. The Questionnaire

The response rate for the parental questionnaire was 97.4% (76 out of 78). The majority (71.1%) of the parents stated that their children carry the schoolbag to school alone, and (85.3%) also stated that their child does not leave the bag at school, but rather carries it to and from school each day.

### 3.2. Descriptive Statistics 

The participation rate for anthropometric and kinematic measurements was 92.3% (72 out of 76). Table 1 displays the anthropometric values of height, mass, Body Mass Index (BMI) and the mass of a full schoolbag. The majority (80.6%) of children were within normal mass parameters relative to their height, or in other words between the 5th and 85th percentile according to the WHO (World Health Organization) growth table [23]. The average mass of a full schoolbag was 4.51 kg, and the mass of a full schoolbag relative to the average mass of the subjects was 16.11%. Table 2 displays the average values the craniovertebral, craniocervical and anterior pelvic tilt angle unencumbered and encumbered. All postural variables (CV, CC and APT) were within normal parameters although review of the literature has shown that there is a high degree of variability in said parameters [24,25,26,27,28,29].

### 3.3. COG, Craniovertebral, Craniocervical and Anterior Pelvic Tilt Shift

A statistically significant shift of center of gravity due to carrying a schoolbag in all projections and along all axes (*p* < 0.01) was found (Table 3).

The greatest average shift of center of gravity occurred in the sagittal axis (2.4 cm shift forward). The average vertical shift of center of gravity was 0.7 cm in the downward direction. The center of gravity also shifted along the coronal axis an average amount of 0.7 cm to the left. The CV and CC angles showed statistically significant changes (*p* < 0.01). The CV angle value was reduced by 5.2° on average, while the CC angle value increased by 2.5° on average under the mass of a schoolbag (Table 3). The only parameter not displaying any statistically significant changes was the APT angle (*p* > 0.05). Since the largest shift of center of gravity occurred along the sagittal axis, it is possible that leaning forward is an important factor in postural readjustment to carrying a schoolbag, which led us to test the correlation between a shift of center of gravity with statistically significant changes in the head and neck. Correlations with anthropometric factors such as mass, height and BMI were also calculated, in order to ascertain their significance (Table 4).

Body mass showed the highest negative correlation with the shift of center of gravity along the sagittal axis (*p* < 0.01), while height also showed a significant negative correlation (*p* < 0.01). The BMI showed a small negative correlation (*p* < 0.05), the CC angle a very small and slight negative correlation (*p* = 0.46), while the CV angle change did not prove to be statistically significant (*p* > 0.05). As the APT angle change was not statistically significant, we did not expect any significant correlation, which proved to be correct (*p* > 0.05). The final analysis was an exploration of the difference in the shift of center of gravity and neck angles between boys and girls (Table 5).

Contrary to expectation, a statistically significant difference was found in the shift of center of gravity along the vertical axis. COG for male children shifted inferiorly 0.8 cm, while inferior shift for female children was 0.5 cm (*p* < 0.05). All the other parameters proved to be statistically insignificant.

## 4. Discussion

### 4.1. Shift of Center of Gravity

The average ratio of schoolbag to subject mass in this study was 16.11%, which matches the upper limit of the average ratio in other studies [2,25]. Average height, mass and BMI were typical of 7-year-old children. The shift of center of gravity while the subjects were encumbered with a schoolbag was tested, and the anterior shift concurs with the results of many other studies [30,31,32,33] which measured the anterior shift of the torso in a standing position or during bipedal locomotion. Since the anterior shift of center of gravity while under a load in the form of a schoolbag is a well-known fact, the primary goal of this study was to measure the extent of the shift. A comprehensive study from 2005 [11] was conducted on a slightly older sample of 10- to 15-year-old children. That study used kinematic and kinetic parameters to ascertain the shift of center of gravity and support center, which yielded an average anterior shift of 3.4 cm. In testing the shift of center of gravity, they used a load amounting to 20% of the subject’s mass, which is higher than the average load used in this study. This leads us to the logical conclusion that a greater load ratio contributes to a greater anterior shift of center of gravity, as several studies confirm such a conclusion [11,31,33,34,35]. Interestingly enough, the same study calculated a significant superior shift of center of gravity by 0.2 cm, which is contrary to the results of this study, which ascertained an inferior shift by 0.7 cm. One possible explanation is that the difference in results of the two studies could be caused by the age difference of the subjects. Subjects in the current study were seven years of age, while the mean age of subjects in study by Talbott [11] was 12.4 years. The age difference could explain different findings because older children could have better postural control, even though they were carrying a heavier load. For example, one study [32] reviewed literature on trunk backward and forward lean and found that younger children had more pronounced lean forward than older children when encumbered with the same load ratio relative to the child’s mass. The authors hypothesized that an undeveloped musculoskeletal system was more prone to postural compensations when encumbered to better regain balanced position (center of gravity above center of base of support). Even though the seven year olds in our study experienced lesser anterior COG shift, they found greater balance by shifting inferiorly as well as anteriorly. Unfortunately, literature comparing the same relative load in children of different ages is lacking, and more research is needed. It is important to note that carrying a bag on one’s back shifts COG superiorly and posteriorly as a result of added mass [11]. This is caused by COG of the load being in a posterior and superior position relative to the body’s center of gravity. The body compensates for this posture by shifting COG anteriorly and inferiorly, thus securing a stable and balanced posture. Another interesting point is the shift of center of gravity along the coronal axis by 0.7 cm to the left relative to the subject. Since a single schoolbag was used in the study, it can be assumed that the bag’s center of gravity was not exactly in the middle, i.e., that the bag was not uniformly filled (the right side perhaps being heavier), which could have caused the shift of body’s center of gravity to the left as a compensatory action due to the position of the schoolbag on the right side of the center of gravity. A uniformly filled bag would secure the same position of the center of gravity along the coronal axis as that of the body, and the shift most likely would not occur. Since no studies were found which measured the lateral shift of center of gravity, it is not possible to compare the above mentioned data, but it is possible to interpret several studies [34,36] that researched the link between an asymmetrically burdened spine and posture while carrying a schoolbag. All three studies point to a similar conclusion that a symmetric load causes fewer postural adaptations than an asymmetric one. Two further points that occur due to a change of center of gravity: (i) when the body is under a load, the center of gravity will not return to the same position it had before the load was applied, regardless of the postural compensations [35]; and (ii) the body’s center of gravity will remain within the surface of the base when under load, due to normal equilibrium reactions, thus securing stability. In case the center of gravity shifts, a compensatory change in posture will occur in order to return the center of gravity back to approximately the same position. This will produce an uneven distribution of forces in the joints, which can, along with the increased expenditure of energy on actively maintaining posture and in case of a prolonged effect, be a factor in the development of possible musculoskeletal disorders [37]. Shift in COG does not describe in what way the posture has changed, only that masses of body segments have rearranged themselves in a way that body COG has shifted. For that reason, in our study, we have included three postural angles to measure, not only the extent of influence of schoolbag mass on COG, but also the way carrying schoolbag affects posture since those compensatory changes produce uneven distribution of force in joints.

### 4.2. A Change in Postural Angles

A reduction of the CV angle was not surprising, as it confirmed an already established effect [13,38,39] of carrying a bag on the back. The increase in the CC angle cannot be compared to other studies, as no such studies were found. It has to be noted that a reduction of the CV angle can occur due to the flexion of the head and neck, not only due to head protraction [28], which makes the CC angle a more accurate indicator of the degree of head protraction, since the CC angle is reduced in case of head and neck flexion, while it increases in case of head protraction. By comparing the change in CV and CC angles, it is evident that one of the postural adaptations is head protraction. While cause for occurrence of musculoskeletal disorders of the neck is still unclear, there is some evidence that protracted head posture might be positively correlated with pain and disability [22,28,40]. Interestingly, the CV angle did not display a statistically significant correlation with the anterior shift of center of gravity, while the CC angle showed a slight correlation. This is somewhat unexpected, because forward head posture is usually associated with forward trunk lean and anterior COG shift [37]. Although the APT angle did not change significantly, there was a tendency toward an increase of the APT angle, which is logical, since a compensatory anterior shift of the torso occurs when the body is under load. It should also be pointed out that if an increase in the APT angle occurs in individual cases, regardless of the link between anterior pelvic tilt and an increase in lumbar lordosis [26], this likely will not increase lumbar lordosis due to the flexion of the torso during the compensatory anterior shift of the center of gravity.

### 4.3. The Correlation of Mass, Height and BMI with a Change of Center of Gravity

A negative correlation between a sagittal shift of center of gravity and body mass, and a slight negative correlation between a shift of center of gravity and BMI were found. The results indicate that heavier subjects adjust their posture less compared to lighter ones, i.e., the schoolbag represents a smaller percentage of their body mass. A positive correlation between height and the shift of center of gravity was expected, but the analysis produced an opposite result, with height showing a significant negative correlation. A high degree of correlation between mass and height in this study means that a taller person will have higher body mass. The theory of positive correlation between height and center of gravity is based on the assumption that a center of gravity further away from the base of the body is more influenced by changes from external factors, but the results seem to indicate that mass plays a greater role than height. Since the BMI is a value gained from the ratio of mass and height squared, it comes as no surprise that the BMI also has a negative correlation, though a weak one, to the change of center of gravity along the sagittal axis. It can be hypothesized from purely biomechanical angle that heavier and shorter children will suffer less from external influences that destabilize them and influence the change in center of gravity [41]. However, Talbott et al. [11] indicated that overweight children (BMI > 24) had a larger excursion of center of pressure (COP). The authors of the study did not provide a precise explanation on the cause of that phenomenon, but they believed there was a link between proprioceptive issues and obesity. While negative correlation between anterior COG shift and BMI was observed, this might not be the case in overweight children due to neuromotor factors [42]. It should be noted that despite being differently measured, COG and COP in largely static tasks are highly comparable [43].

### 4.4. Differences between Boys and Girls 

The hypothesis that there are no statistically significant gender-based differences is based on two kinematic studies [7,31], which found no differences between male and female subjects. However, one has to take both those studies with caution relative to this one, due to the age difference of the subjects. In our study, boys had larger inferior COG shift. The differences in vertical COG shift found can be explained by biomechanical and neuromotor factors. A study [44] that evaluated posture in 1147 girls and 1266 boys analyzed differences in sagittal plane morphology between genders, showing gender-specific biomechanical loads during habitual upright position at age seven. While gender differences in posture also reflect on COG movement as a part of a postural compensation strategy, the link between postural sexual dimorphism and COG shifts has yet to be explained by future studies. Other possible explanation for difference in vertical COG shift is that girls have better postural stability than boys. One study [45] has found evidence that there are differences in lower path velocity, smaller radial displacement and lower area velocity of center of pressure, which indicate better postural stability. Since inferior movement of COG is compensation for schoolbag load, girls have smaller COG shift because they have better postural stability.

### 4.5. Limitations of this Study

The primary limitation of this study was the inability to use a highly accurate 3D optoelectronic kinematic system and complex biomechanical modeling as well as an inverse dynamics approach that would give us more information about the forces and moments in the joints. While this approach enables more detailed data acquisition and analysis, it also increases time to process the signal and significantly increases financial resources needed. Likewise, the use of pressure plates in this study would have enabled the tracking of the center of pressure during measurements, and simultaneous compare changes in center of gravity and center of pressure, which could have produced higher quality results. In addition, a larger sample size with a larger age range would allow for more reliable and generalizable results and conclusions of this study. 

## 5. Conclusions

This study has shown that an anterior shift in center of gravity of the body and a change in neck and head angles occur in first-grade primary school students when carrying a load in the form of a schoolbag weighing 4.51 kg. It also appears that heavier children experience a smaller shift in center of gravity when under the same load as children with lower body mass. The posture measured, while carrying a schoolbag, has been related to musculoskeletal disorders of the neck [22,40,46], but since children spend a relatively low period of time carrying the bag it is impossible to ascertain whether carrying a schoolbag leads to chronic disorders of the musculoskeletal system.

### Suggestions for Further Studies

Perrone et al. mentioned in their conclusion [4] that much of the previous research is limited to biomechanical laboratories, and that more studies should be done in real-world setting using contemporary loads that students carry in real-life situations, so that the impact of these loads can be measured, as well as their potential biomechanical and physiological effects on students’ bodies. This study aimed to address shortcomings stemming from purely laboratory-based research; however, more similar studies are needed to draw definitive conclusions. 

Likewise, almost all previous studies were based exclusively on a postural model, which was used to explain the occurrence of musculoskeletal pain in youth, which in turn served as basis for the prediction of possible chronic musculoskeletal disorders. Numerous studies have shown that there is no clear correlation between the mass of a schoolbag and poor posture, and developing chronic musculoskeletal disorders as a consequence [27,34,42,47,48]. Furthermore, an ongoing discussion is currently trying to determine the importance of the postural–structural–biomechanical model [49,50,51], and the possible consewquent appearance of chronic musculoskeletal disorders. The appearance of lower back pain (LBP) is a case in which psychosomatic influences are stated as possible factors in the development of chronic LBP [48]. Clinical experience often shows a difference between the subjective state and objective case presentation. It must therefore be stressed that the biomechanical approach to the issue is just one possible avenue to take when trying to solve this issue, as studies conducted so far have not provided an unequivocal solution to the issue, and that other approaches, such as psychosocial, and ergonomic ones, should be pursued. 

In addition, future research should aim to incorporate state-of-the-art equipment into the study and should include more respondents and age groups.

## Figures and Tables

**Figure 1 ijerph-16-03946-f001:**
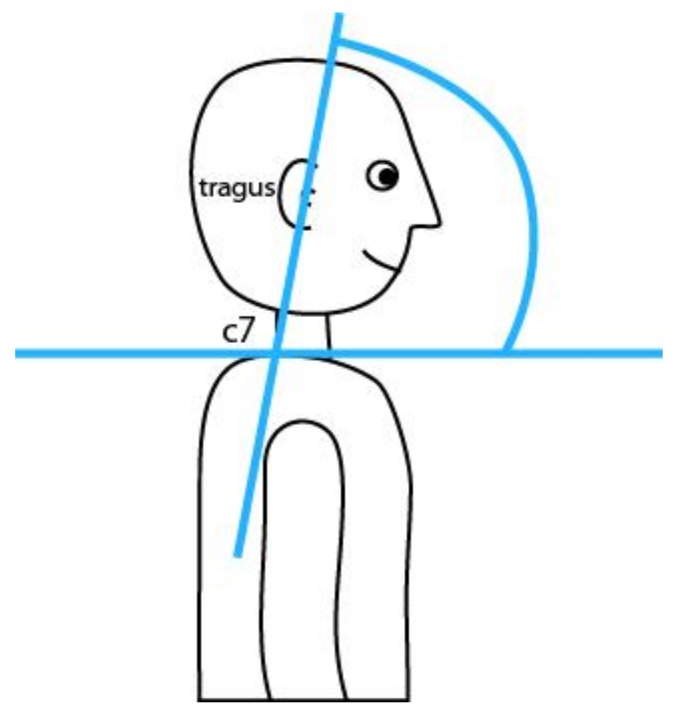
Illustration of craniovertebral angle.

**Figure 2 ijerph-16-03946-f002:**
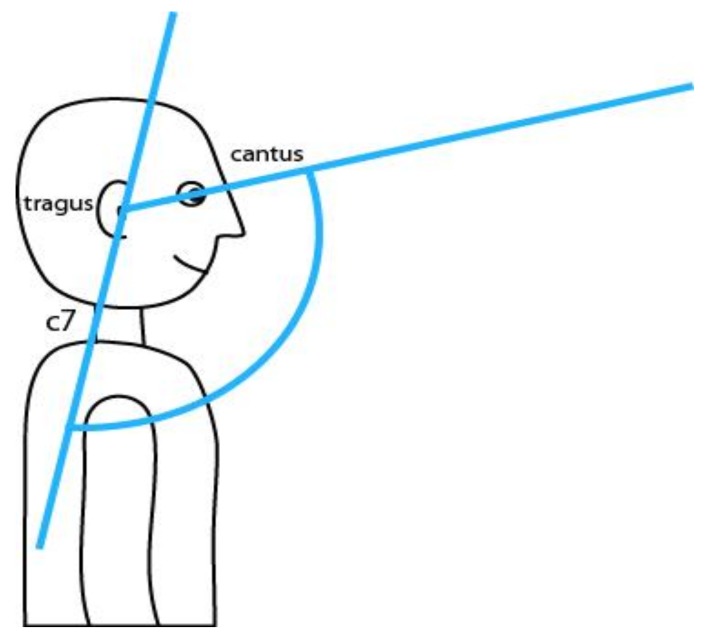
Illustration of craniocervical angle.

**Table 1 ijerph-16-03946-t001:** Basic anthropometric parameters.

Variables	Mean	SD	Min	Max	n
Height (cm)	129.3	6.2	118.8	148	72
Mass (kg)	27.9	5.6	18.7	45.1	72
Schoolbag mass (kg)	4.51	1.09	2.7	7.12	63
BMI	16.66	2.6	12.47	24.8	72

**Table 2 ijerph-16-03946-t002:** Postural angles without and with schoolbag.

Variables	Mean	SD	Min	Max	n
CV without (°)	55.7	4.5	44.7	68.2	61
CC without (°)	136.2	7.7	116.7	156.2	61
APT without (°)	17.7	8.8	1.12	40.01	45
CV with (°)	50.6	4,9	35.8	61.1	61
CC with (°)	138.8	9.2	114.3	167.9	61
APT with (°)	18.7	8.1	5.4	35.6	45

Legend: CV- craniovertebral angle, CC- craniocervical angle, APT- anterior pelvic tilt angle.

**Table 3 ijerph-16-03946-t003:** Paired samples *t*-test of COG and postural angles shift with mean difference.

Variables	ΔMean	SEM	t	*p*	n
Lateral COG shift	−0.7	1.2	−4.95	<0.01**	71
Vertical COG shift	0.7	0.6	9.84	<0.01**	71
Anterior COG shift	−2.4	1.7	−11.7	<0.01**	71
CV (°)	5.2	4.01	9.96	<0.01**	60
CC (°)	−2.5	6.8	−2.76	<0.01**	60
APT (°)	−1.03	9.2	−0.75	>0.05	44

Legend: **p* < 0.05; ***p* < 0.01.

**Table 4 ijerph-16-03946-t004:** Correlation of anthropometric and kinematic parameters with anterior COG shift.

Variables	Pearson r	*p*	n
Mass (kg)	−0.4	<0.01**	71
Height (cm)	−0.4	<0.01**	71
BMI	−0.3	<0.05*	71
CV (°)	0.2	>0.05	60
CC (°)	−0.3	<0.05*	60
APT (°)	0.1	>0.05	44

Legend: **p* < 0.05; ***p* < 0.01.

**Table 5 ijerph-16-03946-t005:** Independent samples *t*-test of COG shift and CV and CC angles shift in boys and girls.

Variables	t	*p*	ΔMean	SEM	n
Lateral COG shift	0.278	>0.05	−0.3	0.3	71
Vertical COG shift	−2.312	<0.05	−0.3	0.1	71
Anterior COG shift	0.732	>0.05	0.3	0.4	71
CV(°)	1.082	>0.05	1.1	1.03	60
CC(°)	−0.467	>0.05	−0.8	1.8	60

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
