# Peer review of "The Influence of the Schoolbag on Standing Posture of First-Year Elementary School Students"

_ijerph, 2019, doi:10.3390/ijerph16203946_

Round 1

Reviewer 1 Report

Accept  after  minor  revisions. See  attached  file (Chool children review.docx

Reviewer 2 Report

Review

Article: Correlation between Schoolbag Mass and Posture in First Year Primary School Students

Title

“Correlation between Schoolbag Mass and Posture in First Year Primary School Students”

I can’t understand the title because no such correlation was investigated. If this was the objective of the study, then it was missed. Every child performed the tests with the same backpack, so this correlation was impossible to be study. 

I propose the following approach: The influence of the schoolbag on static posture of first year primary school students.

Line 9

“The purpose of this study is to”…

“The purpose of this study was to”…

“The purpose of this study is to determine the influence of mass of a schoolbag on standing posture in first year elementary school children”…

On disagree with the present title. But correct with the study content. The title should be retrieved from this sentence, as suggested above.

Line 10

The purpose of this study is to determine the influence of mass of a schoolbag on standing posture in first year elementary school children, and measure the change of center of gravity (COG) as well as change in postural angles of head, neck and pelvis.

…”measure the change of center of gravity (COG) as well as change in postural angles of head, neck and pelvis” are the indicators of posture. I suggest erasing this part of the sentence. So: The purpose of this study is to determine the influence of mass of a schoolbag on standing posture in first year elementary school children.

Line 17

“Analyses of COG position and postural angles have shown that when 1st year primary students are encumbered with a schoolbag (weighing 16.11% of their body mass) their COG shifts, on average, 2.4 cm”.

It was already exposed, on line 16, that the COG shifted 2.4 cm. It seems repeated information.

Line 44

…”as well as lower back pain though to a lesser degree.”

Maybe should be analysed and mentioned the conclusions of more embracing studies. Suggestions:

Spiteri, K., M. L. Busuttil, S. Aquilina, D. Gauci, E. Camilleri and V. Grech (2017). "Schoolbags and back pain in children between 8 and 13 years: a national study." Br J Pain 11(2): 81-86.

Yamato, T. P., C. G. Maher, A. C. Traeger, C. M. Wiliams and S. J. Kamper (2018). "Do schoolbags cause back pain in children and adolescents? A systematic review." Br J Sports Med.

Line 70

“The overall aim of this study is to research the correlation of a schoolbag mass and standing posture in first year elementary school children.”

Considerations about this objective already exposed above.

Methods

Following the idea of a non-laboratory study, testing real conditions, maybe the students should be using their own schoolbags, with their own load. That would have prevented the fallacious lateral COG shift. Furthermore, that change on methodology would have allowed an analysis about a possible correlation between relative (and absolute) schoolbag mass and kinematic variables…

How many sessions were conducted with the same children? All backpacks were weighted in a first session and after the experimental backpack were prepared (with the mean mass) took place the experimental session?

Line 91

“The anthropometric measurements were used to collect basic anthropometric data about the subjects (height, mass).”

Proposal: Subjects (/participants) height and mass was collected.

Line 102

“Anthropometric measurements were measured”…

Anthropometric measurements were taken…

Line 210

What about the correlation about subject’s mass and height with the vertical COG shift? Maybe it could help to explain the difference between girls and boys that was found.

Line 372

Why improve the measurement if it is not clear that those data have practical significance as exposed on lines 249 and 350?

Maybe the future efforts should centre on understand if the time carrying the backpack is enough to lead to chronical disorders as exposed on the conclusion and how could be prevented. Kim, Yi et al. (2008) presented an approach. ("Changes in neck muscle electromyography and forward head posture of children when carrying schoolbags." Ergonomics 51(6): 890-901.)

One more article to compare data, if considered relevant, with participants age starting on 8 years old. (Mosaad, D. M. and A. A. Abdel-Aziem (2015). "Backpack carriage effect on head posture and ground reaction forces in school children." Work 52(1): 203-209.)

Author Response

Please find the authors' response attached.

Reviewer 3 Report

Overall, the study is contributing meaningful knowledge in the correlation between schoolbag mass and posture among young children. The following are a few comments and suggestions for the authors to consider:

Introduction, lines 25-28: Separate this sentence into 2 sentences because it is quite long, “The general public opinion is that schoolbags are too heavy, but science has yet to empirically establish the hazards of carrying a schoolbag to the health of children, the “safe” upper limit of mass for a schoolbag, and how exactly are the mass of a schoolbags and potential musculoskeletal issues connected.”

Introduction: The authors did a great job reviewing the literature leading to the need for this study.

Materials and Methods: The authors provided a clear and detailed description of the instruments used to collect data.

Results: The authors explained the results of the study clearly in the text and tables.

Discussion: Consider including an implication of the study. What would be key takeaways for parents, healthcare providers and educators based on the results of this study?

Round 2

Reviewer 2 Report

English need to be improved to enhance readability.